# Effects of Essential Oils on *Escherichia coli* Inactivation in Cheese as Described by Meta-Regression Modelling

**DOI:** 10.3390/foods9060716

**Published:** 2020-06-02

**Authors:** Beatriz Nunes Silva, Vasco Cadavez, José António Teixeira, Ursula Gonzales-Barron

**Affiliations:** 1Centro de Investigação de Montanha (CIMO), Instituto Politécnico de Bragança, Campus de Santa Apolónia, 5300-253 Bragança, Portugal; beatrizsilva@ceb.uminho.pt (B.N.S.); vcadavez@ipb.pt (V.C.); 2CEB – Centre of Biological Engineering, University of Minho, Campus Gualtar, 4710–057 Braga, Portugal; jateixeira@deb.uminho.pt

**Keywords:** biopreservation, dairy, antimicrobials, mixed-effects model, meta-analysis

## Abstract

The growing intention to replace chemical food preservatives with plant-based antimicrobials that pose lower risks to human health has produced numerous studies describing the bactericidal properties of biopreservatives such as essential oils (EOs) in a variety of products, including cheese. This study aimed to perform a meta-analysis of literature data that could summarize the inactivation of *Escherichia coli* in cheese achieved by added EOs; and compare its inhibitory effectiveness by application method, antimicrobial concentration, and specific antimicrobials. After a systematic review, 362 observations on log reduction data and study characteristics were extracted from 16 studies. The meta-regression model suggested that pathogenic *E. coli* is more resistant to EO action than the non-pathogenic type (*p* < 0.0001), although in both cases the higher the EO dose, the greater the mean log reduction achieved (*p* < 0.0001). It also showed that, among the factual application methods, EOs’ incorporation in films render a steadier inactivation (*p* < 0.0001) than when directly applied to milk or smeared on cheese surface. Lemon balm, sage, shallot, and anise EOs showed the best inhibitory outcomes against the pathogen. The model also revealed the inadequacy of inoculating antimicrobials in cheese purposely grated for performing challenge studies, as this non-realistic application overestimates (*p* < 0.0001) the inhibitory effects of EOs.

## 1. Introduction

Cheeses are ready-to-eat food products that generally are not subjected to any treatment by consumers to ensure their safety before consumption [1]. This product is usually considered safe, due to the physicochemical and antagonistic properties of naturally occurring microflora such as lactic acid bacteria [2]. However, these dairy products can act as vehicles of transmission of foodborne diseases. For instance, in 2017 and 2018, in the European Union (EU), 2.2% and 2.8% of strongly evidenced foodborne outbreaks associated with severe symptoms and a high fatality rate were related to contaminated cheese, respectively [3,4]. Moreover, “milk and milk products” compose 7.7% and 5.4% of the total number of outbreaks reported in 2017 and 2018 [3,4], thus implying that contamination with pathogenic microorganisms does not occur only during cheese manufacturing and that the raw material may be contaminated. When present in milk (raw or pasteurised), pathogenic microorganisms impose a safety issue for cheeses, as bacteria remain viable during long periods of time, even at refrigeration temperatures [5,6,7].

*Escherichia coli*, pathogenic or non-pathogenic, are bacterial agents that can contaminate cheese and have been discussed in numerous studies over the years [1,2,5,8]. Its prevalence in different types of cheese has been studied and described by various authors, who have shown a large range of values: Öksüz et al. (2004) [9] found that 4% of the raw milk cheese samples analysed (n = 50) were contaminated with *E. coli* O157; Stephan et al., (2008) [10] found that 3.7% (n = 432) and 6.3% (n = 364) of the raw milk cheese samples analysed in 2006 and 2007, respectively, were contaminated with STEC (shiga toxin-producing *E. coli*); Rosengren et al., (2010) [8] identified *E.coli* isolates in 34% of the sampled raw milk cheeses (n = 55) and in 3% of cheeses made with pasteurized milk (n = 96). More recently, Ombarak et al., (2016) [11] found *E. coli* prevalence values of 74.5% and 21.7% for Karish (n = 60) and Ras (n = 60) raw milk cheeses, respectively.

Over the past few years, many cheese-associated *E. coli* outbreaks have also been reported [12,13,14,15,16]. According to the most recent EFSA (European Food Safety Authority) and ECDC (European Centre for Disease Prevention and Control) report on zoonoses, STEC was ranked as the fifth most frequent causative agent associated with “milk and milk products” in strongly evidenced foodborne outbreaks in 2018 in the EU (<10%) [4]. Additionally, STEC and “milk and milk products” ranked number 4 in the top 10 pathogen/food vehicle pairs causing the highest number of deaths in strongly evidenced foodborne outbreaks in the EU in 2017 [3].

The prevalence values and several outbreaks reveal the importance of further investigating the contamination of cheese with *E. coli*. Moreover, *E. coli* is particularly concerning as several strains can begin an infection with a small number of cells in the initial inoculum: for example, enteroinvasive and enterohemorrhagic *E. coli* strains (EIEC and EHEC, respectively) may require an infective dose of only about ten cells [17], while the infective dose of *E coli* O157:H7 (STEC) has been estimated to require around 10–100 organisms [8]. To reduce *E. coli* contamination and proliferation in cheese, it is essential to implement measures that can guarantee microbiological safety throughout the manufacturing process, distribution, and storage. One alternative is through the incorporation of antimicrobials in the product, a control measure that is common practice in industry.

A novel approach to antimicrobials has been the replacement of chemicals with plant-based antimicrobials, such as essential oils (EOs) and plant extracts. In the last few years, several researchers have demonstrated the antimicrobial capacity of essential oils (EOs) in several food matrices, including cheese and other dairy products, by performing challenge tests of inoculated pathogenic or non-pathogenic *E.* coli in cheese with added EOs [18,19,20,21,22]. A meta-analysis of published results can synthesise, integrate, and distinguish the outcomes from various studies, producing a more precise estimate of the effect size, with increased statistical power, than is possible with a single study [23]. Meta-analyses are useful in food safety research to address numerous research questions, including the effect of interventions [23]. In this sense, the objective of this research was to provide an insight into the effectiveness of EOs for *E. coli* control in cheese through a meta-regression approach, intended for the optimisation of the use of such biopreservatives to improve the microbiological safety of cheeses. Furthermore, through the construction of a multilevel meta-regression model, the effects of the different EOs, methods of application, antimicrobial concentration, and exposure time can be disentangled and understood.

## 2. Materials and Methods

### 2.1. Data Collection and Description of the Data Set

An electronic, systematic literature search was carried out in Scopus, PubMed and Web of Science databases to find original articles, published since 2000, reporting on the application of EOs in cheese making and their efficiencies against generic or pathogenic *E. coli*. The search aimed to find quality studies validated by the scientific community.

The bibliographic searches were conducted by properly applying the AND and OR logical connectors to combine terms regarding biopreservation and terms referring to biopreservatives’ characteristics and capacities in the selected products as follows: (preservative OR extract OR bio-preservati* OR biopreservati* OR “essential oil”) AND (antimicrobial OR inhibitory OR natural OR plant OR functional) AND (activity OR capacity OR propert* OR effect*) AND (cheese). Grey literature was not acquired to avoid data validity concerns and data duplication, since high-quality theses and reports are likely to be also published in peer-reviewed journals. Other meta-analysis studies and systematic reviews were also excluded. The criteria for inclusion of data were: (i) the temperature of storage and antimicrobial concentration must be reported in each study; (ii) each study must have collected mean log reduction values, at least, at 4 distinct time points (or, alternatively, 4 sets of mean treatment and mean control values, so that the reduction could be calculated); (iii) no mixture of essential oils; (iv) only positive mean log reduction values (no growth data); and (v) if an antimicrobial film was used, the control must also have been coated with the film but without the antimicrobial (as opposed to uncoated).

After assessing all the information from the publications, sixteen studies (N = 362) published from 2000 until August 2019 were considered appropriate for inclusion [18,19,20,21,22,24,25,26,27,28,29,30,31,32,33,34]. From the selected studies, information on the study ID, source of the EO (plant name), strain and/or serotype, mean log reduction, sample size (number of samples used to calculate the mean of the log reduction), storage temperature (°C), exposure time (defined as the time, in days, at which the log microbial reduction was quantified in the challenge study), EO concentration, pathogen inoculum level (log CFU/g or mL), and application type (defined as the mode of application of the antimicrobial; namely, milk, film, cheese surface, and cheese mixture), were collected.

The application type “milk” refers to the direct addition of the antimicrobial agent in bulk milk before curding, whereas the application type “cheese surface” refers to the practice of smearing the cheese surface with the tested antimicrobial. The category “film” was assigned to those challenge studies where the antimicrobial was embedded in the packaging material through micro- or nano-encapsulation. The application type “cheese mixture” was a special category created to accommodate results from those challenge studies whose experimental methodology consisted of grinding cheese, inoculating it with the pathogen, and adding the antimicrobial. Thus, “cheese mixture” does not reflect a real mode of application of antimicrobials in the cheese manufacturing process context, but an experimental protocol for challenge studies that researchers have probably devised for being handy but not realistic. For simplification, the types of application “cheese mixture” and “cheese surface” will be referred to as “mixture” and “surface,” respectively.

Figure 1 and Figure 2 describe the *E. coli* square-root log reduction data retrieved as a function of the square-root of exposure time and of the natural logarithm of the antimicrobial concentration, respectively, for cheeses with essential oils incorporated by distinct application methods. The square-root and natural logarithm transformations were necessary to normalise data distribution. A further description of the data set is summarised in Table 1. Table 2 compiles the study characteristics extracted from each primary study and the distribution of the mean log reduction data among the different levels of the study characteristics extracted.

### 2.2. Meta-Regression Modelling

A mixed-effects linear model with weights was adjusted to the full data set to describe the antimicrobial effect of EOs on the square-root of log reduction (*√R*). According to the information provided on the pathogen strain and/or serotype, a new class variable was defined, “pathogenicity”, composed of the levels “pathogenic” and “non-pathogenic”. Variables or moderators defined for data analysis encompassed application type (*App*), exposure time (*t*), antimicrobial concentration (*Conc*), antimicrobial concentration unit (*ConcUnit*), and pathogenicity (*Pathog*). The variables exposure time and antimicrobial concentration were square-root and natural-logarithm transformed, respectively, to reduce heteroscedasticity. Due to a lack of or uneven data, not all levels could be evaluated in the meta-regression. More specifically, cheese categories described in Table 1, storage temperature, and inoculum level could not be evaluated in the model.

The meta-regression model adjusted to the meta-analytical data was of the form,
(1)$$\sqrt{R_{ikmn}}=\left(\beta_{0}+u_{i}\right)+\beta_{1n}Pathog_{n}+\beta_{2k}App_{k}+\left(\beta_{3k}+v_{i}\right)App_{k}(\sqrt{t)}\;+\beta_{4m}ConcUnit_{m}+\beta_{5m}ConcUnit_{m}\left(LnConc\right)+\varepsilon_{ikmn}$$
where *β*_0_ is an intercept, **β*_1*n*_* and *β*_2*k*_ are the set of fixed effects of the *n* types of pathogenicity (class variable consisting of the levels: pathogenic and non-pathogenic) and of the *k* types of application (class variable consisting of the levels: cheese mixture, cheese surface, milk, and film), respectively; *β*_3*k*_ is a vector representing the effects of the square-root of exposure time *√t* nested within application type, which allows the slopes of exposure time to take different values depending on the type of application *k* used; *β*_4*m*_ is the set of fixed effects of the *m* antimicrobial concentration units (class variable consisting of the levels: %v/v and %w/w), which allows comparing outcomes expressed in different units; and *β*_5*m*_ is a vector describing the impact of the natural logarithm of the antimicrobial concentration (*LnConc*) nested within the antimicrobial concentration unit (*ConcUnit*). *LnConc* and *ConcUnit* were linked in the nested term “*ConcUnit(LnConc)*” to evaluate the EO concentration effect on its own, without the impact of the measure unit and implies that the slopes of antimicrobial concentration could take different values depending on the concentration unit *m*.

The remaining unexplained variability was extracted by placing random-effects *u_i_* due to antimicrobial type *i* in the intercept *β*_0_ and random effects *v_i_* due to antimicrobial type *i* in the square-root of the exposure time slope *β*_3*k*_. These random effects *u_i_*, *v_i_* were assumed to be correlated following a normal distribution with mean zero and a variance-covariance matrix (*s_u_*^2^, *s_uv_*, *s_v_*^2^) from where the correlation coefficient *ρ* of the random effects was calculated. The error term *ε_ikmn_* accounts for the residuals and follows a normal distribution with a mean of zero and a variance of *s*^2^. Model parameters, as affected by moderators, were calculated from the fitted meta-regression, and the significance of moderators was evaluated by an analysis of variance (α = 0.05).

By this random-effects arrangement, it was possible to assess the effectiveness of the EOs by comparing the random effects *u_i_* (intercept) and *v_i_* (EO concentration slope). In this analysis, the EO-specific intercept and slope values are interpreted as deviations *u_i_* and *v_i_* from the mean values *β*_0_ and *β*_3*k*_, respectively. Thus, it was assumed that the higher the *u_i_* and *v_i_*, the stronger the antimicrobial effect of the EO*_i_*.

In order to obtain precise estimates of the antimicrobial effect on pathogen inactivation and to reflect the quality of research design, different weights were assigned to each primary study according to the sample size (*n*) used along the experiment to evaluate microbial inactivation. When a source did not present the number of replicates sampled to calculate the pathogen reduction, *n* = 3 was assigned, as this was the modal value in the database.

To evaluate the fraction of variability in *√R* that could be explained by the moderators (R^2^), a null model version (no moderators) of Equation (1) was fitted, and τ² was calculated as (*s_u_*^2^ + *s_uv_* + *s_v_*^2^). From the fitted full model (Equation (1)), *τ*^2^*_res_* was calculated as (*s_u_*^2^ + *s_uv_* + *s_v_*^2^), and finally R^2^ was estimated as (*τ*^2^
*−*
*τ*^2^*_res_*)/*τ*^2^. The histogram of Pearson’s residuals (estimates of experimental error calculated from the difference between the observed values and the predicted values) was also produced to verify the robustness of the model. The meta-regression model described was fitted using the *lme* (linear mixed-effects models) function from the *nlme* package implemented in R software (version 3.6.2, R Foundation for Statistical Computing, Vienna, Austria) [35].

## 3. Results and Discussion

The results of the analysis of variance of the meta-regression adjusted are presented in Table 3. The model allowed for the inclusion of several moderating variables; however, some terms were not included as fixed effects as they were highly confounded with other variables, or because the data was not equally distributed among the different levels of a variable. This was the case for all cheese descriptive categories depicted in Table 1, storage temperature, and inoculum level. Nonetheless, information on the term “Strain/Serotype” was used to group strains/serotypes in two classes, pathogenic and non-pathogenic, thus creating a new moderating variable, “Pathogenicity”.

In Table 3, the significance of the terms “pathogenicity”, “application type”, “App(√t)”, and “ConcUnit(LnConc)” reveals the impact of such variables on the microbial reduction promoted by EOs in cheese. The term “App(√t)” not only shows that exposure time has a strong influence on microbial reduction, but also that such an effect is dependent upon the mode of application of the antimicrobial in cheese. In this sense, this term indicates that some modes of EOs application are more effective than others for pathogen inactivation, and that to achieve a certain microbial reduction, distinct exposure times are needed according to the mode of application selected. The term “ConcUnit” shows the need to properly group outcomes originating from different units, so that a correct evaluation of the results is possible. In that sense, the term “ConcUnit(LnConc)” reveals the positive association between antimicrobial concentration and microbial inhibition, while linking the antimicrobial concentration used with its corresponding unit.

The fitted parameters of the meta-regression modelling the antimicrobial effect of EOs against *E. coli* are presented in Table 4. A clear tendency for microbial reduction is observed when EOs are incorporated in cheese, as revealed by the positive intercept *β*_0_. Further insight on the variables affecting microbial inactivation is provided by an analysis of the remaining parameters. The *β*_1*n*_ values demonstrate the distinct efficiency of EOs depending on the pathogenicity of the targeted bacteria: non-pathogenic *E. coli* is expected to be more susceptible to EO action (*β*_1*n*_ = 0), whereas pathogenic *E. coli* seems to present higher resistance (*β*_1*n*_ = −0.200). The EOs’ inhibitory capacity is generally due to their ability to degrade and damage cellular walls, cell membranes, and membrane proteins, enhancing the cell membrane permeability and leading to the escape of bacterial cell contents [36]. Moreover, EOs have also shown antimicrobial activity against pathogenic *E. coli* due to their inhibitory effects on biofilm formation and on major virulence factors, such as shiga toxin, through the inhibition of shiga toxin encoding genes and through phage induction and production [37,38].

The *β*_2*k*_ values provide further insight into the different microbial reductions achieved when distinct application modes are used, as previously revealed by analysis of variance through the significant term “application type”. In the model, the mode of application “mixture” is considered the “base value”, with a mean of zero (*β*_2*k*_ = 0.000), implying that the remaining application types reflect positive and negative deviations from that mean. Essential oil incorporation within films surrounding the product (*β*_2*k*_ = 0.023) or into cheese mixtures attained an overall lower microbial reduction, while their addition in milk (*β*_2*k*_ = 1.312) revealed to be the application type leading to the highest microbial inactivation, followed by a smearing of the cheese surface (*β*_2*k*_ = 0.642).

From the *β*_3*k*_ parameter, the different mean values indicate that distinct exposure times to the antimicrobial are needed to achieve a target microbial reduction depending on the application type. More specifically, it can be observed that applying the EO to a cheese mixture (*β*_3*k*_ = 0.676) or within a film (*β*_3*k*_ = 0.281) promotes faster inhibitory effects than applying the EO to the cheese surface (*β*_3*k*_ = 0.078) or into the milk (*β*_3*k*_ = 0.075). When comparing the results of the *β*_3*k*_ parameter against those of *β*_2*k*_, it is possible to observe that nesting the square-root of exposure time within the application type leads to different outcomes in terms of antimicrobial action of those application methods. In fact, the *β*_2*k*_ parameter suggests an increase of inhibitory capacity as follows: mixture<film<surface<milk; whereas *β*_3*k*_ suggests the following order for the effect of the antimicrobial in time: milk<surface<film<mixture. Although using the experimental practice of the “cheese mixture” for a challenge study may point out to the highest rates of inactivation, this application method does not represent the real cheese manufacturing process context, and moreover, this inoculation protocol seemingly leads to significantly overestimated values of *E. coli* reduction, showing that this is not a suitable methodology for challenge studies. Instead, internationally accepted guidelines for conducting challenge tests of food products are provided in ISO 20976-1:2019, which recommends test units representative of a food matrix to be: (i) the complete content of the packaging unit; or (ii) aseptically sampled portions from the packaging unit or from the bulk food [39].

In this sense, the results suggest that incorporation of EO into films appears to be the most effective defence against *E. coli*, whereas application in milk or on the surface yields lower, similar, results. In films, the retention and release properties of encapsulated EOs in the polymer matrix are determinant for antimicrobial efficacy, as this mechanism ensures the release of consistently effective inhibitory doses over long periods of time [40]. In milk, however, a slow, controlled release of the EOs is not possible, and, moreover, its antimicrobial activity is impaired by milk fat content [41] and other food components. These issues may justify the lower efficacy of EOs in milk matrix, compared to films, observed in our study.

The mean values of *β*_4*m*_ show a divergence of the inhibitory effect depending on the antimicrobial concentration unit used. The significance of this parameter highlights the importance of properly grouping antimicrobial concentration values according to its units for an appropriate assessment of the results. For this reason, the two variables LnConc and ConcUnit were linked in the term “ConcUnit(LnConc)”, in order to evaluate the EO concentration effect on its own, without the impact of the measure unit, so that more precise estimates (reduced standard errors) could be obtained. The positive intercepts of the term “ConcUnit(LnConc)”, described by *β*_5*m*_, depict the positive association between antimicrobial concentration and microbial inhibition, meaning that higher EO concentrations lead to greater inhibitory effects. The similar mean values of *β*_5*m*_ reveal that *E. coli* inactivation is reasonably the same regardless of the antimicrobial concentration unit used (%v/v or %w/w).

The analysis of random-effect marginal intercepts and natural logarithm of antimicrobial concentration slopes is presented in Table 5. Very distinct antimicrobial effects can be achieved depending on the selected EO, as shown by the large variability of the intercept and slope values. Although all the EOs meta-analysed presented bactericidal effects against *E. coli*, assessing the random effects, it seems that lemon balm, sage, shallot, and anise present the greatest bactericidal effects against *E. coli* in cheese. 

The outcomes of another meta-analysis study aiming to describe *L. monocytogenes* and *S. aureus* inactivation by essential oils [42] also revealed the high-level antimicrobial effects of lemon balm and sage EOs against those pathogens. The similar results from our investigation demonstrate the efficacy of the EOs from these two specific plants against the growth of both Gram-positive (*L. monocytogenes* and *S. aureus*) and Gram-negative (*E. coli*) bacteria. However, available literature suggests that Gram-negative bacteria are generally more resistant to EOs than Gram-positive bacteria, due to different cell membrane compositions [36], so it is possible that the antimicrobial efficacy of lemon balm and sage against *E. coli* is reduced in comparison to their efficacy against *L. monocytogenes* and *S. aureus*, even if lemon balm and sage EOs were the ones revealing the greatest inhibitory action against *E. coli* out of all the EOs retrieved. Nonetheless, it is important to refer to a recent study focusing on polyphenols, a class of plant secondary metabolites highly responsible for antibacterial activity [43], that showed that microbial sensitivity to antimicrobial agents is, among several factors, strain-dependent [44]. This supports the need for strain as a characterization level when evaluating the antimicrobial effectiveness of biopreservatives such as EOs.

The essential oils described in Table 5 can be attributed to five plant taxonomic families: *Apiaceae* (anise), *Ranunculaceae* (black cumin seed), *Lamiaceae* (lemon balm, oregano, rosemary, sage, thyme, *Zataria multiflora* Boiss.), *Amaryllidaceae* (shallot) and *Asteraceae* (tarragon) [45]. The EOs revealing greatest bactericidal effects in our study belonged to the *Apiaceae*, *Lamiaceae*, and *Amaryllidaceae* families. In the last few decades, studies have revealed the broad spectrum of biological activities of the *Apiaceae* species and have shown that both extracts from these plants and isolated groups of compounds (such as phthalides), exhibit antimicrobial properties [46]. Numerous studies have also reported promising results regarding the biological activities of *Lamiaceae*’s compounds, which are predominantly polyphenols and can be found in large quantities [47,48]. Since there is a positive correlation between the antibacterial activities and phenolic content of dietary spices and medicinal herbs [49], EOs of this family could be anticipated as high antimicrobial action promoters. Other studies have also focused on the antibacterial activity of *Amaryllidaceae* plants and its characteristic isoquinoline alkaloid constituents that provide a unique chemical fingerprint for *Amaryllidaceae* plants and are used for antimicrobial screening measures [50].

Nevertheless, it is important that researchers and the food industry also consider the bioavailability of essential oils’ compounds and the EO-food interaction. In this sense, the results of the meta-regression presented here are specific for cheeses only, meaning that they may not be accurate if extrapolated to other food products. The results might be particularly adequate for conclusions on soft cheeses, as this cheese type comprised the highest amount of data (N = 171; Table 1) when compared to hard or semi-hard cheeses (N = 29 and N = 10, respectively; Table 1). Since cheese-related foodborne illnesses have been generally linked to soft cheese or cheese made from raw or unpasteurized milk and rarely hard cheese [2], it was expected that most research available is on soft cheeses.

Heterogeneity analysis was then conducted and revealed that the moderators introduced to the model explained more than 95% of the between-EO variability in microbial log reductions, which is mainly due to the inclusion of moderators that were able to account for the between-EO variability in the log reductions gathered from the literature. This result shows that the differences in microbial reductions were not only due to the EO source but also due to the distinct application types, antimicrobial concentrations, exposure times, and microbial pathogenicity.

Lastly, to evaluate the quality and robustness of the meta-regression model, the histogram of Pearson’s residuals was built, and the goodness-of-fit was assessed, as shown in Figure 3 and Figure 4. The residuals can be considered as elements of variation unexplained by the fitted model, which are expected to be roughly normal and approximately independently distributed, with a mean of 0 and constant variance. The histogram is a fast, graphical method to evaluate residuals, and as seen from Figure 3, this meta-regression model has its residuals symmetrically distributed around zero. Regarding the correlation value of the goodness-of-fit, R = 0.943 can be considered high for a meta-analysis study. In this sense, the histogram and goodness-of-fit support the robustness of the model and its usefulness in providing a valuable insight on the effectiveness of EOs as affected by different variables against *E. coli* growth.

Overall, our research shows that meta-regression modelling may help obtain a greater understanding on the main variables influencing microbial reduction when biopreservatives such as EOs are to be included in fermented dairy products, which is crucial information for the food industry. Moreover, this information can be useful for the experimental design of challenge studies and for the optimisation of the use of EOs as a biopreservation technology for *E. coli* control in cheeses to ensure the microbial quality and safety of this product.

## 4. Conclusions

Literature data was used to build a meta-analytical regression model that summarises the reduction of *E. coli* in cheese achieved by an incorporation of essential oils and clarifies the inhibitory effectiveness by distinct antimicrobials, antimicrobial concentrations, and application methods. 

This meta-regression model showed that the effectiveness of added EOs were regulated by *E. coli* pathogenicity, exposure time, antimicrobial concentration, and the method of application of the biopreservative (cheese mixture, cheese surface, incorporated in film or directly added to milk). The model evidenced that, for a given EO, distinct application methods would require different exposures times to achieve the same microbial reduction. In general, comparing among the factual methods of an application of antimicrobials, EOs’ incorporation in films seem to produce faster *E. coli* inactivation than an application of them onto the cheese surface or in milk. Lemon balm, sage, shallot, and anise EOs showed the highest bactericidal effect outcomes against *E. coli*.

This meta-analysis study also uncovered one issue related to the experimental design of challenge tests. From the results, it can be stated that an incorporation of the antimicrobial in grated cheese (“cheese mixture”) is not an adequate practice for challenge studies, as it does not accurately represent the real manufacturing process and tends to overestimate EOs’ capacity to inactivate pathogens. Instead, approved protocols must be used to investigate how food antimicrobials affect microbial kinetics in challenge studies.

## Figures and Tables

**Figure 1 foods-09-00716-f001:**
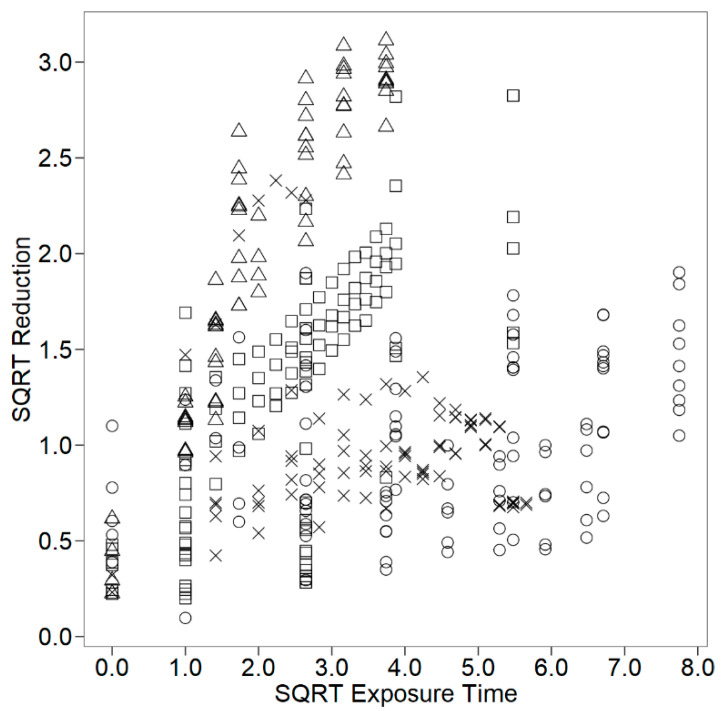
Square-root (SQRT) of log reduction (log CFU/g or log CFU/mL) of *E. coli* as a function of the square-root (SQRT) of exposure time (day) in cheese with essential oils incorporated: in films (□); in milk (○); in cheese mixture (△); on cheese surface (×).

**Figure 2 foods-09-00716-f002:**
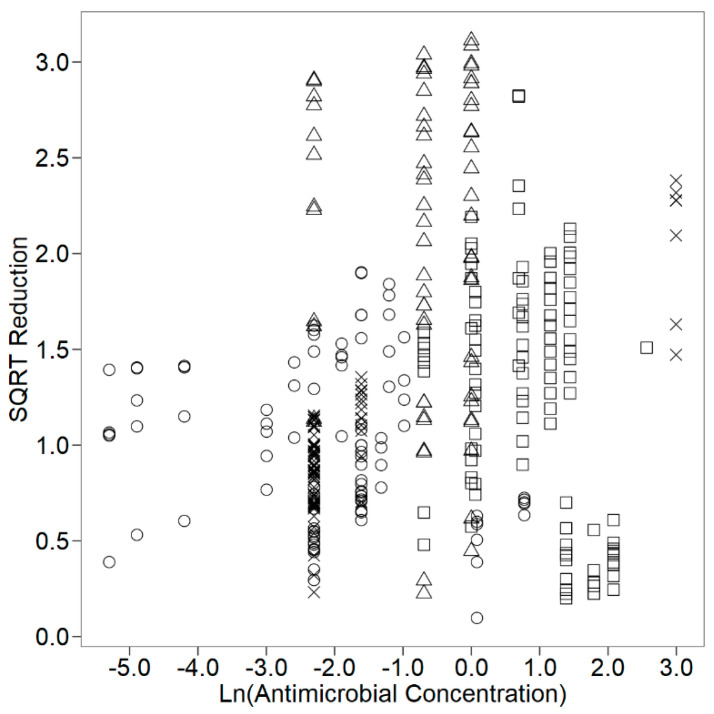
Square-root (SQRT) of log reduction (log CFU/g or log CFU/mL) of *E. coli* as a function of ln(antimicrobial concentration) (%v/v or %w/w) in cheese with essential oils incorporated: in films (□); in milk (○); in cheese mixture (△); on cheese surface (×).

**Figure 3 foods-09-00716-f003:**
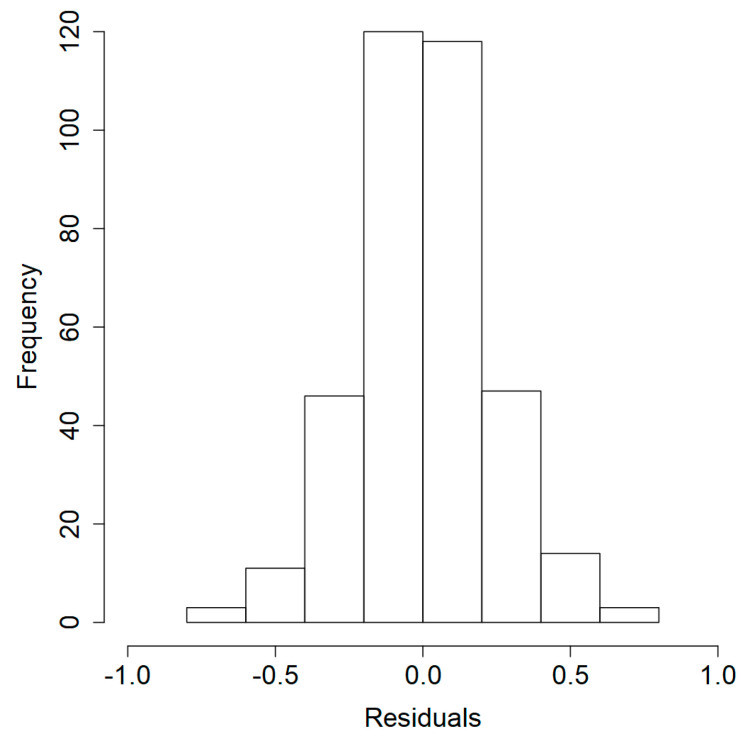
Histogram of Pearson’s residuals of the meta-regression model predicting the square-root of log reduction (log CFU/g or log CFU/mL) of *E. coli* in cheese with incorporated essential oils.

**Figure 4 foods-09-00716-f004:**
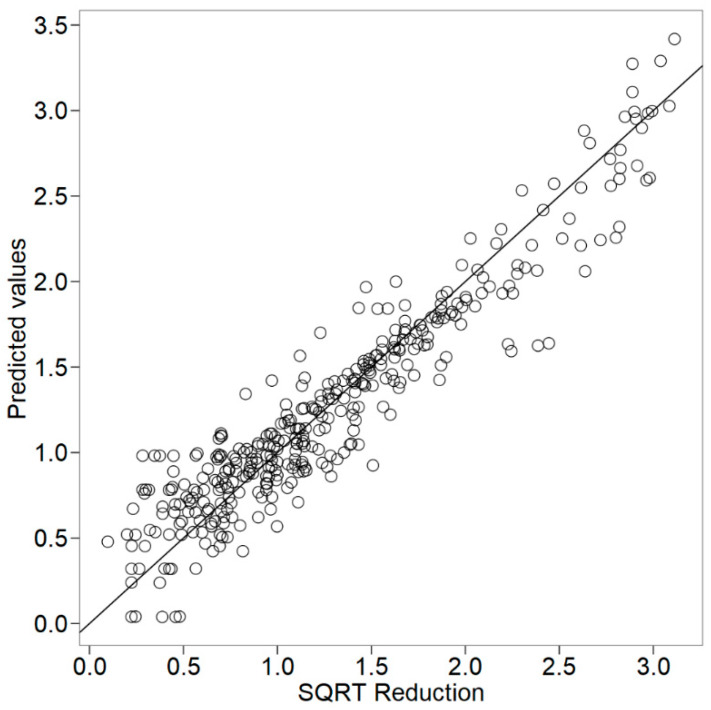
Goodness-of-fit of the meta-regression model predicting the square-root of log reduction (log CFU/g or log CFU/mL) of *E. coli* (R = 0.943) in cheese with incorporated essential oils.

**Table 1 foods-09-00716-t001:** Distribution of log reduction data in *E. coli* by cheese descriptive category for the biopreservatives meta-analyzed.

Categories	Level	N
**Milk treatment**	Pasteurised	147
	Raw	27
	Sterilised	10
	Not stated	178
**Milk species**	Bovine	74
	Caprine	160
	Ovine	29
	Not stated	99
**Type of cheese**	Hard cheese	29
	Semi-hard cheese	10
	Soft cheese	171
	Not stated	152
**Label**	Coalho cheese	7
	Domiati cheese	24
	Feta cheese	160
	Iranian white cheese	62
	Kashar cheese	3
	Lor cheese	24
	Zamorano cheese	29
	Undefined cheese	53
**Starters**	Present	78
	Absent	112
	Not stated	172
**Essay type**	Inoculated	350
	Non-inoculated	12
**Strain/Serotype**	ATCC 8739	24
	CECT 101	1
	EC 16	7
	O157	15
	O157:H7	70
	O157:H7 ATCC 43895	24
	O157:H7 CECT 5947	12
	O157:H7 EDL-932	42
	O157:H7 LH1	34
	O157:H7 M364VO	17
	O157:H7 VT7 negative	56
	PTCC 1533	8
	Not stated	52

**Table 2 foods-09-00716-t002:** Distribution of log reduction data (N) in *E. coli* by moderator for the biopreservatives meta-analyzed.

Moderators	Level	N
**Antimicrobial**	Anise	13
	Black cumin seed	47
	Lemon balm	12
	Oregano	97
	Rosemary	27
	Sage	26
	Shallot	15
	Tarragon	8
	Thyme	46
	*Zataria multiflora* Boiss.	71
**Application type**	Cheese mixture	68
	Film	113
	Milk	98
	Cheese surface	83
**Exposure time, *t* (days)**	0 ≤ *t* < 20 20 ≤ *t* < 4040 ≤ *t* ≤ 60	272 64 26
**Storage temperature, *T* (°C)**	3 ≤ *T* < 1313 ≤ *T* < 2323 ≤ *T* ≤ 35	337 17 8
**Inoculum level, *Inoc*** **(log CFU/g or log CFU/mL)**	1.5 ≤ *Inoc* < 4.254.25 ≤ *Inoc* ≤ 7Non-inoculated	248 102 12
**Antimicrobial concentration, *Conc*** **(%v/v or w/w)**	5 × 10^−3^ ≤ *Conc* < 7 7 ≤ *Conc* < 1414 ≤ *Conc* ≤ 20	343 12 7

**Table 3 foods-09-00716-t003:** Test of fixed effects of the meta-regression models predicting the square-root of log reduction (log CFU/g or mL) of *E. coli* in cheese with incorporated essential oils (EOs) as a function of moderating variables.

Fixed Effects	Num DF/Den DF	*F*-Value	Pr > *F*
Pathogenicity	1/341	84.45	<0.0001
Application type	3/341	41.77	<0.0001
App(√t)	4/341	179.3	<0.0001
ConcUnit	1/341	12.70	0.0004
ConcUnit(LnConc)	2/341	6.291	0.0021

* Num DF and Den DF refer to the numerator and denominator degrees of freedom for the *F*-test (*F*-value), respectively. Pr > *F* is the p-value associated with the *F* statistic of a given fixed effect.

**Table 4 foods-09-00716-t004:** Parameter estimates of the meta-regression model predicting the square-root of log reduction (log CFU/g or mL) of *E. coli* in cheese with incorporated EOs as a function of moderating variables.

Parameters	Mean	SE	Pr > |t|	Heterogeneity
Predictors of √R_ijk_				*τ*^2^*_res_* = 0.316 R^2^ > 95%
*β*_0_ (intercept)	0.661	0.227	0.004	
*β*_1*n*_ (Pathogenicity)				
Non-pathogenic	0	-	-	
Pathogenic	−0.200	0.052	0.000	
*β*_2*k*_ (Application type)				
Application type: mixture	0	-	-	
Application type: film	0.023	0.117	0.842	
Application type: milk	1.312	0.284	0.000	
Application type: surface	0.645	0.200	0.001	
*β*_3*k*_ (App(√t))				
Application: mixture	0.676	0.033	0.000	
Application: film	0.281	0.019	0.000	
Application: milk	0.075	0.012	0.000	
Application: surface	0.078	0.018	0.000	
*β*_4*m*_ (ConcUnit)				
Unit: %v/v	0	-	-	
Unit: %w/w	−0.248	0.112	0.028	
*β*_5*m*_ (ConcUnit(LnConc))				
Unit: %v/v	0.324	0.094	0.001	
Unit: %w/w	0.302	0.093	0.001	
Variances				
*s_u_*	0.562			
*s_v_*	0.260			
*ρ*(*s_u_s_v_*)	0.628			
s (residual)	0.120			

**Table 5 foods-09-00716-t005:** Random effects of the meta-regression models predicting the square-root of log reduction (log CFU/g or mL) of *E. coli* in cheese with incorporated essential oils.

Essential Oil	Intercept	Slope
Anise	**0.087**	**0.025**
Black cumin seed	−1.368	−0.345
Lemon balm	**0.229**	**0.269**
Oregano	−0.193	−0.304
Rosemary	0.228	−0.139
Sage	**0.283**	**0.345**
Shallot	**0.615**	**0.182**
Tarragon	−0.127	0.194
Thyme	0.118	−0.116
*Zataria multiflora* Boiss.	0.128	−0.111

(*) Values in bold highlight the EOs leading to the greatest pathogen inhibition.

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
