# Peer review of "Effects of Essential Oils on Escherichia coli Inactivation in Cheese as Described by Meta-Regression Modelling"

_foods, 2020, doi:10.3390/foods9060716_

Round 1

Reviewer 1 Report

Effects of Essential Oils on Escherichia coli Inactivation in Cheese as Described by Meta- Regression Modelling

Comments to manuscript

This study was aimed to perform a meta-analysis on the effect of literature data to describe the effect of EOs on the inactivation of E coli in a variety of cheeses. The authors have made a meta-regression model with random effects suggesting that the moderators: pathogenicity, together with the application mode, exposure time, and concentration of the antimicrobial exerted the greatest effect on E coli reduction in cheeses. Model’s results have been compared and discussed on the magnitude of such effects on microbial inactivation. The paper is overall very well written and presented, and the methodology is considered appropriate. Though plenty of studies about meta-analysis studies are published in literature, there is a lack of specific information on the effect of EOs in cheese products so that the novelty is justified.

There are some minor comments to be addressed before acceptance:

L31: Introduction can be extended with some more information about the relevance of E coli in cheeses (prevalence, concentration levels found, no. outbreaks etc). It is surprising that E coli has been the addressed pathogen instead of Listeria, which is the most representative microbial hazard in cheese products.

L61: I think it can be replaced by exposure time, which was the assessed variable

L68: this is somehow difficult to evaluate. Have the authors taken some criteria for the selection of the studies?

L125: Please explain this further, which variables were not evaluated?

L136: It should be explained more clearly why the term ‘conc unit’ was left as a fixed effect in beta 4, and nested with LnConc in beta 5?

L154: Why n=3 was left as a default value? There could be some studies with a lower number of replicates

L165: It would be good to show in Table 3 (if possible) the non-significant terms too. Besides, the column Num/Den DF should be defined

L179: Please explain which application modes are the best ones according to the model's results
L228-230: But their effect largely depends on the exposure time... Could the authors indicate the minimum time that exerted an effect on E coli reduction, according to the studied levels (0-20, 20-40, 40-60d)?

L249: It is assumed that those EOs in bold in Table 5 represent the significance of these values

L252: E coli in italics

L307: a reference could be inserted here

L324-325: It sounds contradictory since previously it was demonstrated that application of EOs in milk could produce a lower microbial inactivation in comparison with films.

Reviewer 2 Report

The paper “Effects of Essential Oils on Escherichia coli Inactivation in Cheese as Described by Meta-3 Regression Modelling” addresses a topic worthy of investigation; however, there a is a main drawback, which limits the significance of the study:

the authors searched for the antimicrobial activity and standardized the results as Square Root of Log Reduction. However, they did not consider that the antimicrobial activity relies upon the protocol of EOs application. I can see that the authors consider this variable (milk, cheese surface, film…), but there is another information required: the effective concentration of EOs, which is different from the amount added by researchers in the mixture or in a film. The effective concentration is that actually active on the targets.

I suggest to include this variable in the analysis and see what happens. Probably, some outcomes change.

Another challenge: it is not correct to group strains as pathogens or not-pathogens. Each strain should be considered alone and after that it is important to see if they show different trends or can be grouped in class. If authors want to perform a preliminary grouping, the most correct one is the categorization of E. coli in the five serotypes (EAEC, EHEC, EPEC, EIEC, ETEC).

Finally, I can see a short discussion…I suggest a focus on the practical implication of a meta-analysis and on its importance for industry…

Round 2

Reviewer 2 Report

The authors addressed my issues; the paper can be accepted for publication